# OpenReview forum: "JailBreakV: A Benchmark for Assessing the Robustness of MultiModal Large Language Models against Jailbreak Attacks"
_colmweb.org/COLM/2024/Conference — COLM_

### Official Review · Reviewer_xKBf · 2024-05-11

**Rating:** 6
**Confidence:** 3
**Ethics Flag:** 1

**Summary:**

The authors investigate whether the jailbreaks that are effective in eliciting problematic responses from LLMs, also transfer to multimodal LLMs. To this end, the authors build upon an existing benchmark and expand it to include similar unsafe prompts (textual expansion), and various related/unrelated images (visual expansion), providing a multimodal benchmark comprising 28k examples. The authors find that the prompts that elicit unsafe responses from LLMs, also transfer to multimodal LLMs. Furthermore, these unsafe prompts are effective regardless of the image input associated with them – whether related or not. This work adds to the existing volume of work on safety evaluations on LLMs, and to this end presents interesting insights about the transferability of unsafe prompts between LLMs and multimodal LLMs. However, I have some concerns about the motivation and phrasing of the work, as well as the methodological choices of the work.

**Questions To Authors:**

**Q1:** Could the authors clarify why the prompt to expand the unsafe prompts instructed the LLM to generate 100 examples at once, instead of generating them one-by-one?
**Q2:** Are there any evaluations that establish the quality of these expanded prompts – perhaps on a subset of the data?

**Reasons To Accept:**

- Evaluating the tendency of multimodal LLMs to respond to unsafe multimodal prompts is an important and timely problem
- The paper is easy to follow, and well-structured
- I particularly liked how the authors expanded the unsafe textual prompts to include related/noisy images to increase the coverage of their experiments

**Reasons To Reject:**

- Major concern: The phrasing of the work is confusing and somewhat inconsistent . The main contributions of the work is to assess the transferability of unsafe textual prompts for LLMs to multimodal LLMs. To this end, the authors construct a synthetic benchmark that adds images to known unsafe prompts. I think the claims (and the title of the paper) should reflect that the benchmark is for evaluating the transferability of unsafe textual prompts, and is not a general-purpose benchmark. This is important because, in principle, multimodal LLMs, have a greater attack/jailbreak “surface”, as the input can span multiple modalities. However, from that broad attack/jailbreak surface, this “benchmark” only evaluates for a very specific, and perhaps, narrow subset of jailbreaks.
- Methodological weakness: Now, moving to the methodology that is adapted to construct the synthetic benchmark. First of all, the textual expansion of unsafe prompts is not IID, as the prompts to generate the unsafe prompts asks the LLM to generate 100 examples at once. Even though the authors have subsequent steps to filter out highly similar unsafe prompts from the generated set, it is unclear why the authors did not generate one unsafe prompt at once, which would have been a close approximation to the IID condition. Furthermore, there is not evaluation of the quality of these expanded unsafe prompts.
- Lastly, as a minor point, while discussing the results pertaining the role of image input, I would suggest the authors to clarify that in situations where the image encoder is not frozen, it is possible to leverage the image input much more prominently to design a jailbreak.

Very minor comment:
- The citations are rendered without parentheses in several places throughout the text

---

> ### Author Rebuttal · Authors · 2024-05-31
>
> # Response to {Reasons To Reject}:
> ## A1: Major concern
> We do not ONLY evaluate the unsafe textual prompts. All the jailbreak samples in our benchmark involve text and image inputs. We also included 8000 unique attack queries based on two attack methods (QR and Figstep), where the images are malicious and the corresponding text is benign.   We believe our benchmark comprehensively evaluates MLLM jailbreak safety from BOTH textual and image attack surfaces.
> ## A2: Methodological weakness
> We want to clarify that we ask LLM to generate 100 unsafe QUESTIONs at once. This is because it will generate more diverse questions. We empirically observe that when data are generated one at a time, LLM tends to produce highly similar data even with varying seeds. We generated 20 questions with GPT4, and only find 5 unique ones. However, ours will increase the diversity. We compared the similar across the different questions. The similar score is 0.18, which is lower than commonly used question sets, e.g., Advbench (0.33). We think the reason is LLMs are autoregressive models so that each question is generated condition on the previously generated questions.
> We used Llama Guard to evaluate the toxicity of the GPT-generated data. The results are shown below:
> | Policy                       | Toxicity (%) |
> |------------------------------|--------------|
> | Animal Abuse                 | 65.2         |
> | Economic Harm                | 98.8         |
> | Fraud                        | 98.8         |
> | Government Decision          | 48.54        |
> | Health Consultation          | 20.99        |
> | Malware                      | 98.85        |
> | Tailored Unlicensed Advice   | 69.84        |
> | Unethical Behavior           | 59.72        |
> | Violence                     | 49.55        |
> | **Average**                  | **69.2**     |
>
> ## A3 : Frozen image encoder
> Our benchmark is designed to evaluate the jailbreak robustness of MLLMs at the inference stage, so the entire model, including the image encoder, is frozen. We believe the image input is fully leveraged to design the jailbreak sample in our benchmark: In our dataset construction process, we have included 8000 unique attack queries based on two state-of-the-art image attack (QR and Figstep),  where the images are malicious and the text is benign. These methods ensure a broad and comprehensive evaluation of MLLMs' vulnerabilities.
> # Response to {Questions To Authors}:
> ## A4: See A2
> ## A5: See A2

---

> > ### Author Response · Authors · 2024-06-04
> > **We will be glad to receive your further feedback**
> >
> > Dear Reviewers,
> > As you might know, we are approaching the discussion period deadline. We deeply appreciate your comments and made detailed responses to all your concerns. If your concerns have been addressed, we sincerely appreciate that you can update the score.
> > Please feel free to reach out in case you need any clarification.
> > Best Regards,
> > Authors.

---

> > ### Comment · Reviewer_xKBf · 2024-06-06
> > **Response to rebuttal**
> >
> > I would like to thank the authors for their response, and for presenting additional results. Regarding the point on generating 100 samples at once, as opposed to one-by-one, I now see the authors' rationale, and would encourage them to include this in the updated version of the paper.
> >
> > Regarding the comprehensiveness of the benchmark, I am still on the fence regarding whether or not to the attacks covered in this work are comprehensive for multimodal models. Multimodal models do have an increased attack surface area due to multiple input streams (covered in this work to some extent), but ALSO because of how data in these input streams interact with each other (unexplored in this work). The benchmark takes existing unimodal attacks and seeming mix-and-matches them to build multimodal attack, without leveraging the interplay that could exist in these modalities. I believe this is a limitation that should be made more prominent among the major claims of this work, and also plays against the novelty of this work.
> >
> > Nonetheless, since the rationale for generating 100 samples at once makes sense, I am increasing my score to reflect that.

---

> ### Author Response · Authors · 2024-06-06
> **Response to reviewer**
>
> Thank you for your feedback and for raising your score based on our rebuttal.
> In our benchmark, we have included the most "interacting" jailbreak methods that were proposed by the time we submitted (Figstep and QR), which leverage both image and text modality simultaneously to jailbreak MLLMs. We will continue to monitor the community for any new multimodal jailbreak methods focusing on interaction in the future. Thanks for your suggestions.
> Best regards, Authors

---

### Official Review · Reviewer_SmWp · 2024-05-12

**Rating:** 7
**Confidence:** 3
**Ethics Flag:** 2

**Summary:**

The paper proposes JailBreakV-28K, a comprehensive dataset consisting of 28K test cases to evaluate the susceptibility of multimodal large language models (MLLMs) to jailbreak attacks from textual and visual inputs. By generating 20,000 text-based prompts that successfully jailbreak LLMs and combining them with different image types, as well as constructing 8,000 image-based jailbreak instances leveraging state-of-the-art techniques, the paper demonstrate the high transferability of LLM jailbreak methods to MLLMs. Extensive experiments on different open-source MLLMs show the significant vulnerability, with text-based attacks proving really effective, regardless of the image content.

**Ethics Concerns Details:**

The benchmark may contain a lot of harmful queries, but this paper lacks discussion on the potential problems of releasing the benchmark.

**Questions To Authors:**

- The majority of current MLLMs are derived from LLM (i.e., build MLLM after the pre-training of LLM such as LLaVa), which suggests that they should be sensitive to LLM's jailbreak techniques (the conclusion of this paper). Have you conducted any comparative analysis on different MLLM solutions to see which technical approach is the most robust against LLM attacks?

**Reasons To Accept:**

- This paper is well written, and its motivation is very clear.
- The JailBreakV-28K benchmark can serve as a valuable resource for researchers and developers working on improving the safety and alignment of MLLMs
- The experimental results obtained using JailBreakV-28K reveal critical vulnerabilities for today's MLLMs. The high success rates of transferred LLM jailbreak attacks, particularly text-based attacks, underscore the pressing need for improved defense mechanisms.
- The paper's insights have practical implications for the development and deployment of safe and robust MLLMs.

**Reasons To Reject:**

- Given the existence of numerous jailbreaking MLLM benchmarks in prior research (Table 1), it appears that consolidating them could potentially create a robust jailbreak benchmark. Therefore, it can be argued that the originality of the benchmark proposed in this paper might not be particularly significant.
- Although the benchmark was proposed to uncover jailbreaking issues, it is crucial to consider the potential problems of releasing the dataset, as it could lead to numerous security challenges. However, I did not find any statement on how to release the dataset in a responsible manner.

---

> ### Author Rebuttal · Authors · 2024-05-31
>
> # Response to {Reasons To Reject}
> ## A1: Potential Lack of Originality
> Thank you for your feedback. However, we believe our benchmark has unique advantages in the comprehensiveness of malicious questions and the originality of MLLMs jailbreak benchmark. Specifically: Table 1 compares RedTeam-2K with other malicious question datasets used in various benchmarks. RedTeam-2K stands out as the largest and most diverse dataset considering both the size and diversity of the data involved. It's essential to note that all the datasets listed in Table 1 primarily focus on LLM jailbreaks, as they serve as benchmarks for MLLM jailbreaks. Regarding the originality of our work, we propose two datasets: RedTeam-2K and JailBreakV-28K. RedTeam-2K is indeed the most diverse and high-quality malicious question dataset available to date. JailBreakV-28K is the first benchmark specifically designed for evaluating MLLM jailbreak protocols, built upon RedTeam-2K. Our contribution lies in introducing the first MLLM jailbreak benchmark to address the growing need for assessing MLLM vulnerabilities. We hope this clarification addresses your concerns regarding the originality of our benchmark.
> ## A2: Responsibility Release of Dataset
> Thank you for your concern. We plan to make the dataset and evaluation code publicly available as an open-source resource. We will take serious action to address safety concerns. Specifically, we will adopt a safety approach to obtain the whole benchmark by conducting a questionnaire and granting permission to researchers who wish to use our benchmark. This will ensure that our benchmark can be used safely.
> # Response to {Questions To Authors}
> ## A3: Comparative Analysis of MLLM Technical Approaches
> We have done evaluations on two types of MLLM solutions: finetuning full LLM backbone and using an adapter module. Unfortunately, the results show that both of the solutions are not robust to LLM’s jailbreak techniques.  We will include more analysis in the future.

---

> > ### Comment · Reviewer_SmWp · 2024-06-03
> > **Thanks for you clarification**
> >
> > Thanks for the rebuttal. I would like to raise my score based on the clarified novelty.
> >
> > Best,
> > Reviewer

---

> > > ### Author Response · Authors · 2024-06-04
> > > **Response to Reviewer SmWp**
> > >
> > > Thank you for your feedback and for raising your score based on our rebuttal. We appreciate your time and consideration.
> > > Best regards,
> > > Authors

---

### Official Review · Reviewer_smBy · 2024-05-12

**Rating:** 7
**Confidence:** 4
**Ethics Flag:** 2

**Summary:**

This work collects various high-quality jailbreak attack examples from existing works and generates a larger machine-crafted example set to enlarge the JailBreak datasets. Also, they incorporate images into several examples and create a new multimodal dataset for safety research. During the process of collecting/generating data examples, the authors provide several heuristic rules and deduplication methods for improving the quality of the dataset.  Based on the generated dataset, authors find that the MLLMs do inherit the vulnerability to specific jailbreak examples from the LLM counterparts.

**Ethics Concerns Details:**

Generate Jailbreak data examples may help misuse open-source LLMs

**Questions To Authors:**

How long does it take for each generation steps take? Please list the GPU type, clear time schedule for evaluating the cost of each step for this data generation process.

**Reasons To Accept:**

1. Research about inherited vulnerability between MLLMs and their LLM counterparts is interesting.
2. Authors made complex data generation and filter pipelines for getting high-quality datasets on jailbreak.
3. A set of fully model-generated jailbreak examples is an interesting dataset.

**Reasons To Reject:**

The llama-Guard-based jailbreak evaluation method may introduce many biases. A more complex evaluation method may give further possibilities.

---

> ### Author Rebuttal · Authors · 2024-05-31
>
> # Response to {Reasons To Reject}
> ## A1: Biased Evaluation Method
> Thanks for the suggestion, in order to make a more comprehensive evaluation, we have added experiments on 2 models for 4 additional evaluation metrics: (i) L means Llama-guard evaluation with default configuration, (ii) H means evaluation metric in Harmbench[1], (iii) LR means Llama-guard evaluation combined with relevant evaluation which instructs Mistral-7B-Instruct-v0.2 to return the relevance of response from MLLMs and the harmful query. and (iv) HR means Harmbench evaluation metric with relevant evaluation. We find that the results are aligned with the conclusions our paper claimed. We will give details of the settings and results in the revised version.
> | Model | (QR)SD | (QR)SD_Typo | (QR)Typo | Figstep |  LLM transfer attack | Evaluation Metric |
> | -------- | -------- | -------- | -------- | -------- | -------- |  -------- |
> | OmniLMM-12B | 21.15    | 33      |  28.26     | 30.05     |  71.39    | L     |
> | OmniLMM-12B | 9.95     | 28.4    |  27.70     | 40.15     |  65.73    | H     |
> | OmniLMM-12B | 7.74     | 22.2    |  22.52     | 32.30     |  46.98    | LR    |
> | OmniLMM-12B | 5.50     | 16.45   |  15.22     | 19.45     |  41.05    | HR    |
> | LLaVA-1.5-7b | 19.75     | 30.55   |  19.44     | 17.75     |  56.06    | L    |
> | LLaVA-1.5-7b | 13.90     | 35.65   |  13.45     | 27.95     |  60.01    | H    |
> | LLaVA-1.5-7b | 11.75     | 32.15   |  11.88     | 25.95     |  44.60    | LR    |
> | LLaVA-1.5-7b | 10.50     | 21.50   |  10.85     | 15.2     |  33.83    | HR    |
>
> [1]: MAZEIKA, M., PHAN, L., YIN, X., ZOU, A., WANG, Z., MU, N., SAKHAEE, E., LI, N.,
> BASART, S., LI, B., FORSYTH, D., AND HENDRYCKS, D. Harmbench: A standardized
> evaluation framework for automated red teaming and robust refusal, 2024.
> # Response to {Questions To Authors}
> ## A2: Generation Time and Cost Evaluation
> We will provide an off-the-shelf dataset for jailbreak robustness evaluation, so it will not be a problem for the community to take generation costs on evaluation. In terms of our construction cost, to generate malicious questions, we manually instruct GPT to generate questions, which is not costly. To generate jailbreak prompts, our average cost for each prompt is about 1 second for 5 sample. We generate the jailbreak prompt on 4 NVIDIA A100 80GB.

---

### Official Review · Reviewer_4zmJ · 2024-05-14

**Rating:** 7
**Confidence:** 3
**Ethics Flag:** 1

**Summary:**

This work investigates the vulnerabilities of MLLMs towards jailbreak attacks, specially on the transferability of llm jailbreak attacks. To this end, the authors construct a benchmark including (1) RedTeam-2K malicious query set (2) JailbreakV-28K jailbreak prompts, and conduct experiments across 10 MLLMs to assess their robustness against jailbreak attacks. The results indicate that the MLLMs are still quite vulnerable towards LLM based jailbreak methods.

**Questions To Authors:**

1. The paper does not report the variances of ASR and can the authors give some observations about the variances of different jailbreak prompts?
2. Sometimes, the generated output is harmful but can be unrelated to the input. In this case, do the authors treat it as successful or failed attack?
3. Is the high ASR across these MLLMs due to the backbone LLMs not being strongly aligned?
4. The authors use Llama-guard to calculate the ASR. Some policies in this benchmark are not included during the Llama-guard training, such as economic harm and malware. Can the system prompt be simply extended to add new policies? Further explanations on this matter would be beneficial.
5. There are some minor typos:
    - 3.2.1 first line: it should be "the first stage."
    - The citation format is incorrect and inconsistent.
    - The authors should include a citation for Llama-guard.
    - In Table 2, some text is bold but lacks explanations.

**Reasons To Accept:**

The gathered malicious queries and chosen multimodal jailbreak prompts cover a comprehensive range of different safety policies. If the benchmark is made public, they could be a valuable resource to the community.

**Reasons To Reject:**

1. Though this benchmark includes numerous MLLMs, the results on state-of-the-art (SOTA) closed-source models like GPT-4V and Gemini remain intriguing. Could the authors provide some analysis of these models?
2. It's expected that MLLMs with the same LLM backbone would exhibit high vulnerability, but a more thorough analysis could be beneficial. For instance, what about the transferability from different LLM models to different backbone LLM models? Figures 4 and 5 only provide partial insights.
3. The jailbreak methods based on visual input are not fully comprehensive. Some common methods, such as VisualAdv[1], ImageHijack[2], and Compositional[3], are absent without explanations.
4. The availability of the code and evaluation datasets is unclear. Will they be made public?

---

> ### Author Rebuttal · Authors · 2024-05-31
>
> # Response to {Reasons To Reject}
> ## A1: Closed-source models
> We add experiments on Gemini. The results show that our benchmark is still effective on Gemini (QR with 30.30 ASR, FigStep with 31.80 ASR, LLM transfer attack with 68.43 ASR).
> ## A2: Transferability of different backbone LLM models
> We observe that jailbreak prompts trained on a specific LLM don't lead to a higher ASR across various MLLMs, even if they share the same LLM as their backbone. For instance, the ASR of the jailbreak generated by attacking Vicuna-7B is 78.89 evaluated on llava-1.5-7b with Vicuna-7B as backbone, which is lower than the ASR evaluated on Qwen-VL-Chat with Qwen1.5-7B as LLM backbone (91.40).
> ## A3: Image perturbation
> We discussed them only in the related work section since they do not have strong transferability across models which is important for benchmark.  We conducted an experiment on VisualAdv and found the transferability of this method is poor (the jailbreak generated by minigpt4 can only achieve 12.9 ASR on Gemini and 1.6 ASR Qwen ).
> ## A4: Dataset publication
> We will release the dataset and code
> # Response to {Questions To Authors}
> ## A5: Variance of ASR
> Our new experiments show the variance of ASR  for OmniLMM-12B, which is low (0.456 on average among different attacks)
> ## A6: Evaluation criteria for harmful but unrelated generated output
> We added experiments about the topic correlation, and the results are consistent with the conclusions in our paper.  We use Mixtral-7b to test the topic relevant and judge whether the attack was successful.  The ASR is 7.74, 22.2, 22.52, 32.30, 46.98 for different attacks (QR)SD, (QR)SD_Typo, (QR)Typo, Figstep, and LLM transfer attack respectively, which is consistent with the tendency revealed by Llamaguard score in our paper.
> ## A7: High ASR Across MLLM
> As analyzed in Sec.4, fine-tuning backbone LLMs in MLLMs will break the safety alignment of the Backbone LLMs.
> ## A8: Safety Policy Configuration of Llama-guard
> Yes, it can be extended to new policies in our evaluations. We configure the safety policy to strictly align with our safety policies. We evaluate Llama-guard with and without configuration. To test the detection accuracy, we manually crafted a malicious content dataset containing 200 samples in different policies. The accuracy rises from 88.5% to 93% after we configure the system prompt, which means the model effectively adapts to new policies, aligning with the conclusion of the Llama-guard official paper.

---

> > ### Author Response · Authors · 2024-06-04
> > **We will be glad to receive your further feedback**
> >
> > Dear Reviewers,
> > As you might know, we are approaching the discussion period deadline. We deeply appreciate your comments and made detailed responses to all your concerns. If your concerns have been addressed, we sincerely appreciate that you can update the score.
> > Please feel free to reach out in case you need any clarification.
> > Best Regards,
> > Authors.

---

> > > ### Comment · Reviewer_4zmJ · 2024-06-07
> > > **Response to Rebuttal**
> > >
> > > I would like to thank the author for providing the additional information, which addressed my questions. I highly encourage the author to include the new results in the final version. I raised my score to 7.

---

> > > > ### Author Response · Authors · 2024-06-07
> > > > **Respone to Reviewer**
> > > >
> > > > Thank you for your feedback and for raising your score based on our rebuttal. We appreciate your time and consideration.  We will update with new results in the final version.

---

### Decision · Program_Chairs · 2024-07-10

**Decision:**

Accept

**Comment:**

This work investigates the vulnerabilities of MLLMs towards jailbreak attacks, specially on the transferability of LLM jailbreak attacks. The authors have done a good job of rebuttal. After rebuttal, it has received scores of 6777. Overall, all the reviewers are happy about the paper, commenting that the paper is well written, and the JailBreakV-28K benchmark can serve as a valuable resource for future researchers working on this topic. One reviewer had the concern that the phrasing of the work is confusing and somewhat inconsistent, and the rebuttal partially addressed it.

The authors are highly encouraged to release their benchmark as promised. Given all the positive scores, the AC would like to recommend acceptance of the paper by the end.

[ethics comment from PCs] Both ethics reviewers highlight necessity of including ethical discussion section at the end of paper. **Ethics statement is mandatory for next revision of the paper.**